# Novel Chrysin-De-Allyl PAC-1 Hybrid Analogues as Anticancer Compounds: Design, Synthesis, and Biological Evaluation

**DOI:** 10.3390/molecules25133063

**Published:** 2020-07-04

**Authors:** Buthina A. Al-Oudat, Hariteja Ramapuram, Saloni Malla, Suaad A. Audat, Noor Hussein, Jenna M. Len, Shikha Kumari, Mel F. Bedi, Charles R. Ashby, Amit K. Tiwari

**Affiliations:** 1Department of Medicinal Chemistry and Pharmacognosy, Faculty of Pharmacy, Jordan University of Science and Technology, P.O. Box 3030, Irbid 22110, Jordan; baoudat@just.edu.jo; 2Department of Pharmacology and Experimental Therapeutics, College of Pharmacy & Pharmaceutical Sciences, University of Toledo, Toledo, OH 43614, USA; hzr0030@tigermail.auburn.edu (H.R.); saloni.malla@rockets.utoledo.edu (S.M.); noor.hussein@rockets.utoledo.edu (N.H.); jenna.len@rockets.utoledo.edu (J.M.L.); 3Department of Chemistry, College of Science and Arts, Jordan University of Science and Technology, P.O. Box 3030, Irbid 22110, Jordan; saaudat@just.edu.jo; 4Department of Pharmaceutical Sciences, College of Pharmacy, University of Nebraska Medical Center, Omaha, NE 68198, USA; fnu.shikha@unmc.edu; 5Department of Medicinal and Biological Chemistry, College of Pharmacy & Pharmaceutical Sciences, University of Toledo, Toledo, OH 43614, USA; Fernand.Bedi@utoledo.edu; 6Department of Pharmaceutical Sciences, College of Pharmacy & Pharmaceutical Sciences, St. John’s University, Queens, NY 11439, USA; cnsratdoc@optonline.net

**Keywords:** triple-negative breast cancer, cytotoxicity, chrysin analogues, flavonoid, anticancer compounds

## Abstract

New chrysin-De-allyl-Pac-1 hybrid analogues, tethered with variable heterocyclic systems (**4a**–**4o**), were rationally designed and synthesized. The target compounds were screened for in vitro antiproliferative efficacy in the triple-negative breast cancer (TNBC) cell line, MDA-MB-231, and normal human mammary epithelial cells (HMECs). Two compounds, **4g** and **4i**, had the highest efficacy and selectivity towards MDA-MB-231 cells, and thus, were further evaluated by mechanistic experiments. The results indicated that both compounds **4g** and **4i** induced apoptosis by (1) inducing cell cycle arrest at the G2 phase in MDA-MB-231 cells, and (2) activating the intrinsic apoptotic pathways in a concentration-dependent manner. Physicochemical characterizations of these compounds suggested that they can be further optimized as potential anticancer compounds for TNBC cells. Overall, our results suggest that **4g** and **4i** could be suitable leads for developing novel compounds to treat TNBC.

## 1. Introduction

Breast cancer is the second leading cause of death among all cancers affecting women [1]. Triple-negative breast cancer (TNBC) lacks the expression of hormone receptors (estrogen (ER) or progesterone (PR)) and/or human epidermal growth factor receptor 2 (HER2), which are more amenable to targeted therapy [2]. TNBC often presents as a high-grade invasive ductal carcinoma (IDC) in patients, accounting for one-fourth of all breast cancer deaths. Consequently, there is an urgent need for the development of compounds that are efficacious and safe for the treatment of breast cancer. Flavonoids are ubiquitous naturally occurring polyphenolic compounds that are commonly found in fruits and vegetables [3]. Flavonoids comprise several classes of low molecular weight compounds, including flavanones, anthocyanidins, flavonols, flavanols, isoflavones, dihydroflavonols, and flavones [4,5,6]. Chrysin (5, 7-dihydroxyflavone, Figure 1) is a natural flavone found in many plant extracts, such as the blue passionflower, as well as in honey and propolis [7,8]. Chrysin has been reported to have properties, such as antioxidant [9], antihypertensive [10], antibacterial [11], anti-inflammatory [12], antiviral [13], antiallergic [14], antidiabetic [15], anxiolytic [16], and anticancer efficacy [17,18]. The activation of apoptosis plays a major role in producing the anticancer efficacy of chrysin [19,20,21]. Mechanistically, apoptosis involves a cascade of initiator and effector caspases [22]. Among these caspases, caspase-3 and caspase-7 are downstream executioner caspases that play an essential role in inducing apoptosis by cleaving a variety of cellular substrates [23]. Therefore, caspase-dependent apoptosis pathways represent targets for the development of efficacious anticancer drugs. Recently, a number of studies have been done to augment the pharmacological activity of chrysin by producing synthetic analogues [24,25,26,27,28,29]. Moreover, there are studies indicating that chrysin-based compounds have in vitro efficacy in breast cancer cells [30,31,32]. The compound, de-allyl procaspase-activating compound 1 (PAC-1), induces apoptosis in different types of cancer cells by activating caspase-3 and/or caspase-7 [33,34,35,36]. Previously, we reported that molecular hybridization between chrysin and de-allyl PAC-1 can be used to produce novel hybrid molecules with cytotoxic efficacy [29]. Molecular hybridization is a process that comprises the amalgamation of two or more pharmacophoric moieties of different bioactive molecules into a single molecular framework [37,38].

In this study, new chrysin-de-allyl PAC-1 hybrid analogues, substituted with variable aromatic heterocyclic cores, were synthesized and evaluated to identify a more potent bioactive hybrid against breast cancer cells (Figure 1). The in silico parameters of the synthesized compounds were calculated to predict their pharmacokinetic profile and drug-likeness using SwissSimilarity ADME, a web tool [39]. Subsequently, the target compounds were screened for antiproliferative efficacy in the human breast cancer cell line MDA-MB-231, using the MTT 3-(4,5-dimethylthiazol-2-yl)-2,5-Diphenyltetrazolium Bromide) colorimetric assay. Finally, we determined the effects of the most potent compounds on the cell cycle.

## 2. Results and Discussions

### 2.1. Chemistry

The target compounds (**4a**–**4o**) were prepared according to Scheme 1, starting from the commercially available chrysin, using previously published synthetic procedures [29]. Briefly, chrysin was added to methyl 2-bromoacetate at a low temperature in the presence of K_2_CO_3_, yielding the desired alkylation product **2**. The conversion of ester **2** into hydrazide **3** was accomplished using 80% hydrazine hydrate and a few drops of hydrochloric acid at low temperature. ^1^H and ^13^C NMR data confirmed the formation of hydrazide **3** as a pure single product. Target compounds (**4a**–**4o**) were obtained by adding hydrazide **3** to an appropriate aldehyde in the presence of a catalytic amount of hydrochloric acid at room temperature. All analogues gave adequate analytical and spectroscopic data, which were in full accordance with their structures. ^1^H and ^13^C NMR spectra showed that compounds **4a**–**4o** existed as geometrical isomers (E/Z isomers). The E:Z ratio for each compound was determined from ^1^H NMR spectra utilizing the integration of the neat methylene group peaks that appeared as two separated singlets for each isomer. Target compounds were analyzed using SwissSimilarity (Swiss Institute of Bioinformatics, Lausanne, Switzerland). Compounds **4a**–**4j** and **4n**–**4o** were the most structurally similar to apigenin, a flavonoid similar to chrysin. The physicochemical properties of the compounds (**4a**–**4o**) are shown in Table 1.

### 2.2. 3-(4,5-Dimethylthiazol-2-yl)-2,5-Diphenyltetrazolium Bromide-Based Cytotoxicity Assay

The synthesized chrysin derivatives were evaluated for cytotoxic efficacy in the human breast cancer cell line, MDA-MB-231, using the 3-(4,5-dimethylthiazol-2yl)-2,5-diphenyltetrazolium bromide (MTT) colorimetric assay and doxorubicin was used as a reference anticancer drug. The IC_50_ values (concentration of compound in µM required to reduce 50% of cell viability) are shown in Table 2.

As shown in Table 2, the structure–activity relationship (SAR) results indicated that the cytotoxic efficacy of the synthesized compounds was affected by the modifications on the benzene ring (ring D) of the parent compound **4a**. Compound **4a**, possessing a hydroxyl group at C-2 of ring D, had cytotoxic efficacy, with an IC_50_ value of 6.8 µM. To determine the importance of the hydroxyl group at the C-2 position, compound **4b** was synthesized, and its efficacy was similar to compound **4a**. The methylation of the hydroxyl group in **4a** yielded compound **4c**, which had a lower efficacy than **4a** and **4b**, whereas ethylation of the hydroxyl group, which yielded compound **4d,** produced a significant decrease in cytotoxic efficacy, compared to compound **4a**. Previously, we reported that the shifting of the hydroxyl group from C-2 to C-4 (**4e**) produced a significant reduction in the cytotoxic efficacy (IC_50_ ˃ 50 µM). However, methylation of the hydroxyl group in **4e** yielded compound **4f,** which was ~5 times more efficacious than **4e [29]**. The addition of a second hydroxyl group at the C-4 position in **4a** resulted in compound **4g** (IC_50_ = 5.98 µM), which had cytotoxic efficacy similar to compound **4a**. Interestingly, the methylation of the two hydroxyl groups in **4g**, yielding compound **4h,** significantly decreased the cytotoxic efficacy. The addition of a third hydroxyl group to **4g** at position 3 of ring D, yielding compound **4i**, did not significantly alter the antiproliferative efficacy of the compound, whereas adding a third hydroxyl group at position 6 of ring D yielded a totally inactive compound, **4j**. Next, we determined the effect of ring D on the cytotoxic efficacy of the synthesized compounds. Therefore, compounds **4k**–**4m** were synthesized, where ring D was replaced with aromatic heterocyclic moieties. Compounds **4k**, **4l**, and **4m**, containing furan, 3-pyridine, and 2-pyridine moieties, respectively, did not have significant cytotoxic efficacy. Furthermore, adding another fused benzene ring to ring D, while keeping the C-2 hydroxyl group (2-hydroxynaphthalene), yielded the inactive compound, **4n**. The addition of a nitro group at the C5-position of ring D in **4a** yielded compound **4o**, which had an efficacy similar to compound **4a** (IC_50_ = 7.9 µM).

Next, the cytotoxicity and selectivity of the most active compounds, **4g** and **4i**, were determined in a panel of cell lines, including the normal cell lines, HMECs, and cancer cell lines, BT-20, U-251, and HCT116, as shown in Figure 2 and Table 3. Compound **4g** had antiproliferative efficacy in the BT-20, U-251, and HCT116, with IC_50_ values of 5.32, 7.64, and 2.68, respectively. In contrast, compound **4i** had IC_50_ values ranging from 10–25 µM. The results indicated that although compound **4g** is more potent than compound **4i** in the three cancer cell lines, compound **4i** is more selective for the cancer cells than the normal cells.

### 2.3. **4g** and **4i** Induce Apoptosis and G2 Cell Cycle Arrest in a Triple-Negative Breast Cancer Cell Line

Apoptosis, a type of programmed cell death, is one of the major mechanisms by which chemotherapeutic drugs produce their therapeutic efficacy [43]. Morphologically, apoptosis is characterized by cellular shrinkage, which is accompanied with nuclear chromatin condensation and fragmentation followed by blebbing of the plasma membrane. This leads to the formation of small apoptotic bodies that have an intact cellular membrane and unaltered organelle integrity. These bodies are then released in the extracellular environment and removed by the process of phagocytosis [44,45]. Apoptosis can occur by two pathways: The extrinsic pathway and intrinsic pathway. In either pathway, when the cell is exposed to certain extrinsic or intrinsic stimuli, the integrity of the inner mitochondrial membrane of the cell is compromised, resulting in the loss of the mitochondrial membrane potential, and causing the release of several apoptotic factors, including cytochrome c [46,47]. Numerous studies indicate that during apoptosis, phosphatidylserine (PS), in the cytoplasmic side of the plasma membrane, is translocated to the extracellular cell surface [48]. The flipped anionic PS binds to the Ca^2+^-dependent phospholipid-binding protein, annexin V [49]. We discovered and reported several potent apoptosis-inducing compounds [50,51,52,53,54,55,56,57]. In this study, the result of our morphological studies in MDA-MB-231 cells after incubation with our lead compounds, **4g** and **4i,** indicated that apoptosis was occurring (Figure 3A). At a concentration of 20 µM, both compounds decreased the number of adherent MDA-MB-231 cells and induced cellular shrinkage. The cells were rounded and loosely attached, and apoptotic bodies were present. Similarly, the incubation of MDA-MB-231 cells with compounds **4g** or **4i** for 24 h produced a significant loss of the mitochondrial membrane potential. For compound **4g**, the population of cells undergoing apoptosis increased from 14.30% at 0 µM to 26.56% and 58.05% at 5 and 10 µM, respectively (*p* value < 0.0001; Figure 3B,C). Compound **4i** also produced a significant shift in the apoptotic cell population in quadrant II, from 14.78% at 0 µM to 32.25% and 42.56% at 5 and 10 µM, respectively (*p* value < 0.0001, Figure 3B,C). Cell cycle analysis indicated that compounds **4g** and **4i** produced a significant disruption in the cell cycle of MDA-MB-231 cells (Figure 3D). Data obtained using flow cytometry indicated that, when incubated with vehicle alone, MDA-MB-231 cells had a normal cell cycle (5.43%, 83.4%, 4.63%, and 4.46% in the subG1, G1, S, and G2 phases, respectively). However, incubation of MDA-MB-231 cells with compound **4g** resulted in a significant shift towards the G2 phase (59.12% and 54.13% for 5 and 10 µM, respectively (*p* value < 0.0001, Figure 3E).

### 2.4. Compounds **4g** and **4i** Activate Apoptosis by Activating the Intrinsic Apoptotic Pathway

Apoptosis can be induced by the activation of two major pathways: The intrinsic and extrinsic apoptotic pathways [22]. The activation of the intrinsic apoptotic pathway induces the activation of proapoptotic proteins, such as apoptosis regulator Bak (Bcl-2 homologous antagonist/killer) and Bax (Bcl-2-associated X protein) [58]. The activated the Bax and Bak proteins subsequently permeabilize the mitochondrial outer membrane by forming pores on its outer surface [58,59,60]. Consequently, cytochrome c (Cyt C) is released into the cytosol, where it combines with the adaptor protein (Apaf-1) to form an apoptosome [22]. The initiator caspases (i.e., caspase-2, caspase-8, caspase-9, or caspase-10) are activated and recruited to large protein complexes, resulting in the cleavage of the executioner caspases, caspase-3 or caspase-7 [61].

Since MDA-MB-231 cells incubated with compounds **4g** and **4i** had a decrease in the mitochondrial membrane potential, which is an early event of intrinsic apoptosis, i.e., altered permeability of the inner mitochondrial membrane, we conducted experiments to determine if these compounds altered the expression of key apoptotic proteins, including cytochrome c, in MDA-MB-231 cells using Western blotting analysis. Our results indicated that compounds **4g** at 5 µM, and compound **4i** at 10 µM produced a significant increase in the expression of cytochrome c, compared to cells incubated with vehicle (Figure 4A,B). This may be due to an increase in the expression of Bak following incubation with **4g** and **4i** (Figure 4A,B), compared to cells incubated in the absence of lead compounds. In addition, both compounds (**4g** at 5 µM and **4i** at 10 µM) produced significant cleavage of the initiator caspase, caspase 9, in MDA-MB-231 cells, compared to cells incubated with vehicle (Figure 4A,B). These events activated caspase 7 in breast cancer cells incubated with both concentrations of **4g** and 10 µM of **4i**, compared to cells incubated with vehicle (Figure 4A,B). In contrast, there was no significant change in the mammalian target of rapamycin (mTOR) expression, indicating that cell death induced by **4g** and **4i** in MDA-MB-231 cells is not due to autophagy (Figure 4A,B). Thus, our results suggest that cytochrome c release induces intracellular initiator caspase activation, followed by the activation of executioner caspases, thus activating apoptotic cell death machinery through intrinsic the pathway.

## 3. Materials and Method

### 3.1. Chemistry

All chemicals and solvents were procured from commercial sources (reagent grade) and were used without further purification. The reaction progress was monitored by thin layer chromatography (TLC) using precoated TLC plates of silica gel 60 F254. ^1^H-NMR and ^13^C-NMR spectra were recorded using a 400 MHz Bruker Avance Ultrashield spectrometer. The spectra were obtained in ppm using automatic calibration to the residual proton peak of the solvent, dimethyl-sulphoxide (DMSO-d_6_). The ^1^H NMR data are presented as follows: Chemical shift (δ ppm), multiplicity (s = singlet, d = doublet, t = triplet, q = quartet, m = multiplet), coupling constants (Hz), and integration. The ^13^C NMR analyses were reported in terms of the chemical shift. The ^1^H and ^13^C NMR spectra for all compounds are included in the Appendix A. HRMS data were acquired using a Thermo QExactive Plus mass spectrometer equipped with an electrospray ionization source (Thermo Fisher Scientific, Greensboro, NC, USA).

#### 3.1.1. Synthesis of methyl 2-((5-Hydroxy-4-oxo-2-phenyl-4*H*-chromen-7-yl)oxy)acetate (**2**)

The title compound was synthesized as previously described [62].

#### 3.1.2. Synthesis of 2-((5-Hydroxy-4-oxo-2-phenyl-4*H*-chromen-7-yl)oxy)acetohydrazide (**3**)

The title compound was synthesized according to our previously published procedure [29].

#### 3.1.3. General Procedure for the Synthesis of *N*’-arylidene-2-((5-Hydroxy-4-oxo-2-phenyl-4*H*-chromen-7-yl)oxy)acetohydrazide (**4a**–**4o**)

To a stirred suspension of hydrazide **3** (1.0 g, 3.0 mmol) in anhydrous methanol (60 mL), the appropriate aldehyde (3.0 mmol) was added, along with a few drops of concentrated hydrochloric acid. After one hour, the reaction was completed and the formed precipitate was separated by filtration, washed with methanol, and dried in air to give pure compounds in good yields.

##### (*E,Z*)-2-((5-Hydroxy-4-oxo-2-phenyl-4*H*-chromen-7-yl)oxy)-*N*’-(2-hydroxybenzylidene)acetohydrazide (**4a**)

The product was obtained as a pale-yellow powder. Yield (64%). ^1^H NMR (DMSO-*d*_6_) (*E:Z* = 1:1): δ 12.84–12.81 (m, 2H), 11.84 (s, 1H), 11.62 (s, 1H), 10.99 (s, 1H), 10.05 (s, 1H), 8.57 (s, 1H), 8.33 (s, 1H), 8.12–8.10 (d, *J* = 8 Hz, 4H), 7.77–7.55 (m, 8H), 7.31–7.24 (m, 2H), 7.07–6.85 (m, 8H), 6.51–6.42 (m, 2H), 5.31 (s, 2H), 4.87 (s, 2H). ^13^C NMR (DMSO-*d*_6_): δ 182.11, 182.06, 167.94, 164.42, 163.65, 163.61, 163.48, 163.37, 161.17, 161.07, 157.36, 157.22, 156.45, 148.27, 141.63, 132.18, 132.12, 131.59, 131.29, 130.57, 130.54, 129.18, 129.15, 129.11, 126.56, 126.47, 120.01, 119.40, 118.65, 116.39, 116.14, 105.44, 105.39, 105.31, 105.09, 98.81, 98.72, 93.62, 93.46, 66.57, 65.36. HRMS (ESI, *m*/*z*): calculated for C_24_H_19_N_2_O_6_ [M + H]^+^ 431.1237; found 431.1238.

##### (*E,Z*)-*N’*-Benzylidene-2-((5-hydroxy-4-oxo-2-phenyl-4H-chromen-7-yl)oxy)acetohydrazide (**4b**)

The synthesis and full characterization of the title compound have been previously reported [29].

##### (*E,Z*)-2-((5-Hydroxy-4-oxo-2-phenyl-4*H*-chromen-7-yl)oxy)-*N*’-(2-methoxybenzylidene)acetohydrazide (**4c**)

The product was obtained as a yellow powder. Yield (55%). ^1^H NMR (DMSO-*d*_6_) (*E:Z* = 1:0.5): δ 12.84–12.80 (m, 1.5H), 11.64 (s, 1.5H), 8.68 (s, 0.5H), 8.37 (s, 1H), 8.11–8.09 (d, *J* = 8 Hz, 3H), 7.91–7.89 (d, *J* = 8 Hz, 1H), 7.82–7.80 (d, *J* = 8 Hz, 0.5H), 7.63–7.55 (m, 4.5H), 7.44–7.4 (m, 1.5H), 7.12–7.0 (m, 4.5H), 6.88–6.84 (m, 1.5H), 6.5–6.42 (m, 1.5H), 5.32 (s, 2H), 4.82 (s, 1H), 3.86 (s, 4.5H). ^13^C NMR (DMSO-*d*_6_): δ182.09, 182.03, 168.09, 164.43, 163.71, 163.60, 163.48, 163.22, 161.13, 161.05, 157.80, 157.64, 157.21, 143.56, 139.73, 132.15, 132.10, 131.72, 131.48, 130.55, 129.09, 126.45, 125.67, 125.53, 121.94, 120.72, 120.66, 111.84, 111.77, 105.42, 105.31, 105.06, 98.77, 98.70, 93.62, 93.45, 66.65, 65.36, 55.68, 54.87. HRMS (ESI, *m*/*z*): calculated for C_25_H_21_N_2_O_6_ [M + H]^+^ 445.1394; found 445.1392.

##### (*E,Z*)-*N’*-(2-Ethoxybenzylidene)-2-((5-hydroxy-4-oxo-2-phenyl-4*H*-chromen-7-yl)oxy)acetohydrazide (**4d**)

The product was obtained as a light-brown powder. Yield (80%). ^1^H NMR (DMSO-d_6_) (*E:Z* = 1:0.5): δ 12.84–12.80 (m, 1.5H), 11.68–11.62 (m, 1.5H), 8.66 (s, 0.5H), 8.40 (s, 1H), 8.11–8.10 (d, *J* = 4 Hz, 3H), 7.90–7.80 (m, 1.5H), 7.60–7.57 (m, 4.5H), 7.41–7.37 (m, 1.5H), 7.10–6.84 (m, 6H), 6.5–6.42 (m, 1.5H), 5.31 (s, 2H), 4.82 (s, 1H), 4.14–4.09 (q, *J* = 8 Hz, 3H), 1.39–1.35 (t, *J* = 8 Hz, 4.5H).^13^C NMR (DMSO-*d*_6_): δ 182.08, 182.03, 168.03, 164.42, 163.77, 163.57, 163.46, 163.23, 161.14, 161.04, 157.20, 157.13, 156.98, 143.46, 140.01, 132.10, 131.67, 131.42, 130.54, 129.09, 126.45, 125.76, 125.58, 122.14, 122.09, 120.61, 112.75, 105.41, 105.29, 105.05, 98.71, 93.63, 93.45, 66.62, 65.36, 63.80,14.63. HRMS (ESI, *m*/*z*): calculated for C_26_H_23_N_2_O_6_ [M + H]^+^ 459.1550; found 459.1548.

##### (*E,Z*)-2-((5-Hydroxy-4-oxo-2-phenyl-4*H*-chromen-7-yl)oxy)-*N*’-(4-hydroxybenzylidene)acetohydrazide (**3e**)

The synthesis and full characterization of the title compound were previously published [29].

##### (*E,Z*)-2-((5-Hydroxy-4-oxo-2-phenyl-4*H*-chromen-7-yl)oxy)-*N*’-(4-methoxybenzylidene)acetohydrazide (**3f**)

The synthesis and full characterization of the title compound were previously published [29].

##### (*E,Z*)-*N’*-(2,4-Dihydroxybenzylidene)-2-((5-hydroxy-4-oxo-2-phenyl-4*H*-chromen-7-yl)oxy)acetohydrazide (**4g**)

The product was obtained as an off-white powder. Yield (85%). ^1^H NMR (DMSO-d_6_) (*E:Z* = 1:0.8): δ 12.84–12.80 (m, 1.8H), 11.67 (s, 1H), 11.43 (s, 0.8H), 11.15 (s, 1H), 9.98 (s, 1.8H), 9.83 (s, 0.8H), 8.42 (s, 1H), 8.20 (s, 0.8H), 8.12–8.10 (d, *J* = 8 Hz, 3.6H), 7.63–7.53 (m, 6.2H), 7.33–7.31 (d, *J* = 8 Hz, 1H), 7.08–7.06 (d, *J* = 8 Hz, 1.8H), 6.89 (d, *J* = 2 Hz, 1H), 6.8 (d, *J* = 2.4 Hz, 0.8H), 6.5 (d, *J* = 2 Hz, 1H), 6.4 (d, *J* = 2 Hz, 0.8H), 6.36–6.30 (m, 3.6H), 5.26 (s, 1.6H), 4.83 (s, 2H). ^13^C NMR (DMSO-*d*_6_): δ182.10, 182.05, 167.42, 164.44, 163.68, 163.64, 163.50, 162.86, 161.14, 161.04, 160.86, 160.53, 159.35, 158.07, 157.23, 149.28, 142.68, 132.18, 132.12, 131.10, 130.55, 129.15, 129.12, 128.27, 126.47, 111.49, 110.37, 107.84, 107.77, 105.45, 105.37, 105.32, 105.07, 102.60, 102.37, 98.81, 98.72, 93.63, 93.45, 66.59, 65.33. HRMS (ESI, *m*/*z*): calculated for C_24_H_19_N_2_O_7_ [M + H]^+^ 447.1186; found 447.1187.

##### (*E,Z*)-*N’*-(2,4-Dimethoxybenzylidene)-2-((5-hydroxy-4-oxo-2-phenyl-4*H*-chromen-7-yl)oxy)acetohydrazide (**4h**)

The product was obtained as a yellow powder. Yield (96%). ^1^H NMR (DMSO-*d*_6_) (*E:Z* = 1:0.5): δ 12.81 (s, 1.5H), 11.49 (s, 1.5H), 8.58 (s, 0.5H), 8.27 (s, 1H), 8.11–8.10 (m, 3H), 7.84–7.57 (m, 6H), 7.05 (s, 1.5H), 6.88–6.84 (m, 1.5H), 6.63–6.41 (m, 4.5H), 5.29 (s, 2H), 4.79 (s, 1H), 3.86–3.82 (m, 9H). ^13^C NMR (DMSO-*d*_6_): δ182.04, 167.79, 164.43, 163.69, 163.55, 163.44, 162.91, 162.55, 162.33, 161.09, 161.02, 159.19, 159.02, 157.18, 143.62, 139.82, 132.10, 130.53, 129.09, 126.78, 126.43, 114.76, 106.41, 105.38, 105.28, 105.02, 98.68, 98.24, 98.07, 93.59, 93.42, 66.66, 65.35, 55.74, 55.40.HRMS (ESI, *m*/*z*): calculated for C_26_H_23_N_2_O_7_ [M + H]^+^ 475.1499; found 475.1499.

##### (*E,Z*)-2-((5-Hydroxy-4-oxo-2-phenyl-4*H*-chromen-7-yl)oxy)-*N*’-(2,3,4-trihydroxybenzylidene)acetohydrazide (**4i**)

The product was obtained as a yellow powder. Yield (85%). ^1^H NMR (DMSO-*d*_6_) (*E:Z* = 1:0.5): δ 12.84–12.80 (m, 1.5H), 11.71 (s, 1H), 11.47 (s, 0.5H), 11.18 (s, 1H), 9.57–9.51 (m, 1.5H), 9.37 (s, 0.5H), 8.54–8.50 (m, 1.5H), 8.38 (s, 1H), 8.18–8.10 (m, 3.5H), 7.61–7.59 (m, 4.5H), 7.07–7.0 (m, 2H), 6.90–6.78 (m, 2.5H), 6.51–6.38 (m, 3H), 5.26 (s, 1H), 4.85 (s, 2H). ^13^C NMR (DMSO-*d*_6_): δ 182.12, 182.07, 167.34, 164.42, 163.68, 163.54, 162.93, 161.16, 161.07, 157.25, 150.45, 148.91, 148.42, 147.47, 146.65, 144.18, 132.76, 132.71, 132.21, 132.16, 130.56, 129.18, 126.49, 121.11, 118.36, 112.11, 110.70, 107.84, 107.73, 105.46, 105.40, 105.34, 105.10, 98.83, 98.75, 93.65, 93.48, 66.59, 65.32. HRMS (ESI, *m*/*z*): calculated for C_24_H_19_N_2_O_8_ [M + H]^+^ 463.1134; found 463.1133.

##### (*E,Z*)-2-((5-Hydroxy-4-oxo-2-phenyl-4*H*-chromen-7-yl)oxy)-*N*’-(2,4,6-trihydroxybenzylidene)acetohydrazide (**4j**)

The product was obtained as a brown powder. Yield (76%). ^1^H NMR (DMSO-*d*_6_) (*E:Z* = 1:0.2): δ 12.83–12.79 (m, 1.2H), 11.65 (s, 1H), 11.48 (s, 0.2H), 10.99 (s, 2H), 10.32 (s, 0.4H), 9.85 (s, 1.2H), 8.72 (s, 1H), 8.41 (s, 0.2H), 8.09–8.08 (m, 2.4H), 7.59–7.58 (m, 3.6H), 7.04 (m, 1.2H), 6.88–6.84 (m, 1.2H), 6.5–6.42 (m, 1.2H), 5.86–5.83 (m, 2.4H), 5.2 (s, 0.4H), 4.82 (s, 2H). ^13^C NMR (DMSO-d_6_): δ 182.08, 166.56, 164.28, 163.61, 163.51, 162.55, 161.74, 161.46, 161.12, 161.04, 159.73, 159.24, 157.20, 147.32, 144.33, 132.15, 130.54, 129.13, 126.45, 105.43, 105.37, 105.11, 98.85, 98.79, 94.37, 93.61, 93.50, 66.58, 65.26. HRMS (ESI, *m*/*z*): calculated for C_24_H_19_N_2_O_8_ [M + H]^+^ 463.1134; found 463.1134.

##### (*E,Z*)-*N’*-(Furan-2-ylmethylene)-2-((5-hydroxy-4-oxo-2-phenyl-4*H*-chromen-7-yl)oxy)acetohydrazide(**4k**)

The product was obtained as a yellow powder. Yield (87%). ^1^H NMR (DMSO-*d*_6_) (*E:Z* = 1:0.5): δ 12.83–12.79 (m, 1.5H), 11.63–11.57 (m, 1.5H), 8.23 (s, 0.5H), 8.11–8.09 (d, *J* = 8 Hz, 3H), 7.92 (s, 1H), 7.85 (s, 1.5H), 7.63–7.55 (m, 4.5H), 7.06–7.05 (m, 1.5H), 6.95–6.83 (m, 3H), 6.64 (m, 1.5H), 6.49–6.39 (m, 1.5H), 5.24 (s, 2H), 4.83 (s, 1H). ^13^C NMR (DMSO-d_6_): δ182.03, 167.97, 164.34, 163.66, 163.59, 163.46, 163.35, 161.13, 161.02, 157.21, 149.12, 148.94, 145.34, 145.10, 137.86, 134.20, 132.11, 130.54, 129.09, 126.45, 113.95, 113.78, 112.16, 105.42, 105.29, 105.08, 98.70, 93.60, 93.38, 66.65, 65.14.HRMS (ESI, *m*/*z*): calculated for C_22_H_17_N_2_O_6_ [M + H]^+^ 405.1081; found 405.1077.

##### (*E,Z*)-2-((5-Hydroxy-4-oxo-2-phenyl-4*H*-chromen-7-yl)oxy)-*N*’-(pyridin-3-ylmethylene)acetohydrazide (**4l**)

The product was obtained as a yellow powder. Yield (86%). ^1^H NMR (DMSO-d_6_) (*E:Z* = 1:0.5): δ 12.82(bs, 1.5H), 11.98 (bs, 1.5H), 8.43 (s, 0.5H), 8.30–8.28 (d, *J* = 8 Hz, 3H), 8.13–7.96 (m, 7H), 7.62–7.55 (m, 4.5H), 7.07–7.06 (m, 1.5H), 6.88 (s, 1.5H), 6.5–6.45 (m, 1.5H), 5.39 (s, 2H), 4.9 (s, 1H). ^13^C NMR (DMSO-d_6_): δ 182.12, 168.50, 164.42, 163.67, 163.55, 161.08, 157.27, 150.51, 148.52, 145.40, 141.22, 133.75, 132.20, 130.59, 129.96, 129.16, 126.51, 123.94, 105.36, 105.12, 98.76, 93.55, 66.61, 65.40. HRMS (ESI, *m*/*z*): calculated for C_23_H_18_N_3_O_5_ [M + H]^+^ 416.1241; found 416.1240.

##### (*E,Z*)-2-((5-Hydroxy-4-oxo-2-phenyl-4*H*-chromen-7-yl)oxy)-*N*’-(pyridin-2-ylmethylene)acetohydrazide (**4m**)

The product was obtained as a brown powder. Yield (52%). ^1^H NMR (DMSO-*d*_6_) (*E:Z* = 1:0.5): δ 12.84–12.81 (m, 1.5H), 11.89–11.87 (m, 1.5H), 8.61 (m, 1.5H), 8.35 (s, 0.5H), 8.11–8.03 (m, 5H), 7.94–7.87 (m, 2H), 7.62–7.55 (m, 4.5H), 7.44–7.41 (m, 1.5H), 7.07–7.06 (m, 1.5H), 6.88 (m, 1.5H), 6.51–6.45 (m, 1.5H), 5.37 (s, 2H), 4.88 (s, 1H). ^13^C NMR (DMSO-*d*_6_): δ 182.02, 168.45, 164.34, 163.75, 163.65, 163.59, 163.48, 161.13, 161.05, 157.21, 152.93, 152.78, 149.51, 149.44, 148.26, 144.45, 136.86, 136.75, 132.09, 130.53, 129.07, 126.44, 124.52, 124.34, 119.98, 119.90, 105.42, 105.30, 105.09, 98.70, 93.63, 93.48, 66.61, 65.30. HRMS (ESI, *m*/*z*): calculated for C_23_H_18_N_3_O_5_ [M + H]^+^ 416.1241; found 416.1238.

##### (*E,Z*)-2-((5-Hydroxy-4-oxo-2-phenyl-4H-chromen-7-yl)oxy)-*N*’-((2-hydroxynaphthalen-1-yl)methylene)acetohydrazide (**4n**)

The product was obtained as a yellow powder. Yield (54%). ^1^H NMR (DMSO-*d*_6_) (*E:Z* = 1:0.5): δ 12.86–12.81 (m, 1.5H), 12.46 (s, 1H), 11.91 (s, 1H), 11.68 (s, 0.5H), 10.77 (s, 0.5H), 9.40 (s, 1H), 8.89 (s, 0.5H), 8.77–8.75 (d, *J* = 8 Hz, 0.5H), 8.30–8.28 (d, *J* = 8 Hz, 1H), 8.12–8.09 (m, 3H), 7.94–7.84 (m, 3H), 7.63–7.58 (m, 6H), 7.42–7.36 (m, 1.5H), 7.24–7.21 (m, 1.5H), 7.08–7.06 (m, 1.5H), 6.94–6.88 (m, 1.5H), 6.55–6.46 (m, 1.5H), 5.38 (s, 1H), 4.94 (s, 2H). ^13^C NMR (DMSO-*d*_6_): δ182.09, 182.03, 167.52, 164.38, 163.64, 163.54, 163.49, 163.17, 161.17, 161.06, 157.95, 157.23, 156.88, 147.37, 143.29, 132.91, 132.52, 132.16, 132.10, 131.59, 131.25, 130.53, 129.13, 128.89, 128.68, 128.13, 127.80, 127.75, 126.46, 123.54, 123.40, 120.97, 118.75, 118.13, 110.08, 108.45, 105.45, 105.33, 105.10, 98.87, 98.73, 93.68, 93.46, 66.72, 65.59.HRMS (ESI, *m*/*z*): calculated for C_28_H_21_N_2_O_6_ [M + H]^+^ 481.1394; found 481.1394.

##### (*E,Z*)-2-((5-Hydroxy-4-oxo-2-phenyl-4*H*-chromen-7-yl)oxy)-*N*’-((2-hydroxy-5-nitrobenzylidene)acetohydrazide (**4o**)

The product was obtained as a yellow powder. Yield (91%). ^1^H NMR (DMSO-*d*_6_) (*E:Z* = 1:0.75): δ 12.85–12.80 (m, 1.75H), 11.97 (bs, 2H), 11.78 (s, 1.5H), 8.64–8.54 (m, 2.75H), 8.33 (s, 1H), 8.15–8.10 (m, 5.25H), 7.64–7.55 (m, 5.25H), 7.08–7.04 (m, 3.25H), 6.90–6.86 (m, 1.75H), 6.52–6.44 (m, 1.75H), 5.39 (s, 2H), 4.89 (s, 1.5H). ^13^C NMR (DMSO-*d*_6_): δ 182.08, 182.03, 168.28, 164.42, 163.71, 163.61, 163.46, 162.04, 161.14, 161.04, 157.20, 144.30, 139.91, 138.65, 132.16, 132.10, 130.55, 129.12, 129.08, 126.73, 126.45, 123.37, 121.68, 120.98, 119.95, 117.10, 116.72, 105.43, 105.37, 105.30, 105.08, 98.78, 98.73, 93.62, 93.46, 66.53, 65.42. HRMS (ESI, *m*/*z*): calculated for C_24_H_18_N_3_O_8_ [M + H]^+^ 476.1088; found 476.1089.

### 3.2. Biological Studies

#### 3.2.1. Cell Lines and Cell Culture

A panel of cancer cell lines, including breast (MDA-MB-231, BT20), brain (U251), and colon (HCT116), as well as a normal cell line (human mammary epithelial cells: HMECs), were grown as adherent monolayers in flasks with Dulbecco’s Modified Eagle Medium (DMEM), supplemented with 10% fetal bovine serum (FBS) and 1% penicillin and streptomycin in a humidified incubator with 5% CO_2_ at 37 °C.

#### 3.2.2. MTT Assay

The (3-(4,5-dimethylthiazol-2-yl)-2,5-diphenyltetrazolium bromide) (MTT) assay was used to determine the cytotoxicity of the 15 chrysin derivatives in the above-mentioned cell lines. Briefly, cells were harvested with 0.05% trypsin, 2.21 mM ethylenediaminetetraacetic acid (EDTA), 1× from Corning (Corning, NY, USA), and suspended at a final density of 5 × 10^3^ cells/well. Cells were seeded (180 μL/well) into 96-well plates. Initially, four different concentrations of each compound were added to find the compounds with the greatest antiproliferative efficacy in the MDA-MB-231 cell line (0, 1, 10, and 100 μM). Subsequently, eight different concentrations (0.1, 0.3, 1, 3, 10, 30, and 100 μM) of each compound were added to the remaining cell lines mentioned above. After 68 h of incubation, 20 μL of the MTT solution (4 mg/mL) were added to each well, and the plates were incubated for 4 h. This allowed viable cells to biotransform the yellow-colored MTT into dark-blue formazan crystals. Subsequently, the medium was discarded, and 150 μL of DMSO were added to each well to dissolve the formazan crystals. The absorbance was determined at 590 nm using a DTX 880 multimode detector (Beckman Coulter life sciences, IN, USA). The IC_50_ ± SD concentrations were calculated from three experiments performed in triplicate/duplicate. The IC_50_ values were calculated from the cell survival percentages obtained for each compound tested at different concentrations. Similarly, the cytotoxicity of the test compounds was compared to the normal cell line (HMEC).

#### 3.2.3. Cell Cycle, Apoptosis, and Mitochondrial Membrane Potential Analysis

MDA-MB-231 cells were plated into 6-well plates at 2.5 × 10^5^ cells/well. The cells were incubated with 0, 5, or 10 μM of compounds **4g** or **4i** and incubated for 12 h. Next, the cells were trypsinized with 0.05% trypsin, 2.21 mM EDTA, 1×, washed, counted, and resuspended in 1 mL of ice-cold PBS. The cells then were stained with propidium iodide (PI) dye and incubated for at least 15 min. The distribution of the cells in each cell cycle phase for the different concentrations was measured using a BD Accuri™ C6 flow cytometer from BD Biosciences (Becton-Dickinson, San Jose, CA, USA) and analyzed using FCS Express 5 plus De Novo software (Glendale, CA, USA).

MitoTracker Red and Alexa Fluor 488 annexin V kits for flow cytometry (Molecular Probes Inc., Invitrogen, Eugene, OR) were used to measure the mitochondrial membrane potential and apoptosis in MDA-MB-231 cells. Briefly, cells were seeded into 6-well plates and incubated with 0, 5, or 10 μM of compounds **4g** and **4i** for 12 h. The cells were then lysed using 0.05% trypsin, 2.21 mM EDTA, 1×, counted, and 4 μL of 10 μM of the MitoTracker Red working solution were added to 1 mL of the harvested cells. The cells were incubated at 37 °C with 5% CO_2_ for 30 min. The cells were washed once with PBS and resuspended in 100 μL of the annexin binding buffer. The cell suspensions were incubated with 5 μL of Alexa Fluor 488 annexin V for 15 min. This was followed by the addition of 400 μL of the annexin-binding buffer. Finally, flow cytometry was used to detect the fluorescence of stained cells at the following excitation/emission maxima: Alexa Fluor^®^ 488 annexin V: 499/521 nm; MitoTracker^®^ Red: 579/599 nm with the BD Accuri™ C6 flow cytometer from BD Biosciences (Becton-Dickinson, San Jose, CA, USA) and analyzed using FCS express 5 plus De Novo software (Glendale, CA, USA).

#### 3.2.4. Protein Expression Analysis Using Western Blot

To measure the expression of Bak, cytochrome c, caspase-7, caspase-9, and mTOR, Western blotting was performed by lysing MDA-MB-231 cells using a lysis buffer (50 mM Tris–HCl, 150 mM NaCl, 1 mM EDTA, 0.5% NP-40, 1% Triton, 0.1% SDS) containing a protease inhibitor cocktail that consisted of Aprotinin, Bestatin, E-64, Leupeptin, and Pepstatin A (Sigma-Aldrich Life Science, St. Louis, MO, USA). The bicinchoninic acid (BCA) quantification assay was used to determine the protein levels in the cell extracts (G-BIOSCIENCES, St. Louis, MO, USA). The extracted proteins were loaded onto a 10–20% tris-glycine gel. After separation, the proteins were transferred from the gel onto a polyvinylidene difluoride (PVDF) membrane. The membranes were blocked using 5% milk in Tris-buffered saline Tween 20 for 30 min and incubated overnight with primary antibodies against Bak (1:1000), cytochrome C (1:1000), caspase-3 (1:1000), caspase 7 (1:1000), caspase 9 (1:1000), mTOR (1:1000), or B-actin (1:2000) in 5% BSA (bovine serum albumin) at 4 °C. The next day, membranes were washed and incubated with horseradish peroxidase-labelled (HRP) anti-rabbit secondary antibody (1:5000 dilutions). The membrane was incubated with the antibody for an additional 1 h. Subsequently, the membranes were washed and developed by Clarity Western ECL substrate (Bio-Rad; Hercules, CA, USA). Protein was detected using a ChemiDoc Imaging System (Bio-Rad). Densitometry analyses of the blots for the detected protein were quantified using the ImageJ software. Data was calculated as ratios of protein/β-actin.

## 4. Conclusions

In conclusion, a series of novel chrysin derivatives were designed, synthesized, and characterized. Upon screening these compounds for their antiproliferative efficacy in the TNBC cell line, MDA-MB-231, and normal breast HMEC cells, two compounds, **4g** and **4i**, had the highest efficacy and selectivity towards MDA-MB-231 cells. Upon investigating the mechanism by which these compounds produce cytotoxicity, it was determined that **4g** and **4i** cause the death of MDA-MB-231 cells by inducing apoptosis, producing cell cycle arrest at the G2 phase, and activating the intrinsic apoptotic pathway. Physicochemical characterizations of these compounds suggested that they can be further optimized as potential anticancer compounds for TNBC cells. The compounds were determined to have some solubility issues that need to be overcome in the future design of additional compounds. Overall, our results suggest that **4g** and **4i** could be suitable leads for developing novel compounds to treat TNBC.

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
