# Peer review of "Novel Chrysin-De-Allyl PAC-1 Hybrid Analogues as Anticancer Compounds: Design, Synthesis, and Biological Evaluation"

_molecules, 2020, doi:10.3390/molecules25133063_

Round 1
Reviewer 1 Report
Novel chrysin-De-allyl PAC-1 hybrid analogues as anticancer compounds: Design, synthesis and biological evaluation
Buthina A. Al-Oudat et al
This manuscript describes the design, synthesis, and biological evaluation of chrysin-desallyl PAC-1 hybrid compounds as anticancer candidates. Although the design and synthesis methods are not novel, and the biological activity of the hybrid compounds is not more potent than that in previous works, but half of novel hybrid compounds have moderate antiproliferative activity against MDA-MB-231, especially, 4e and 4g have higher cytotoxicity to cancer cells than normal cell lines. In addition, the authors also conducted flow cytometry and protein expression studies on 4e and 4g, and confirmed that intrinsic apoptosis leads to cytotoxicity. These results and conclusions will provide Molecules’ readers with interesting information. Therefore, I recommend accepting it for publication. However, the author needs to address some issues.
- Based on Figure 4, compound 4e does not appear to activate Bak in a concentration-dependent manner.
- References 31 and 31 are improperly cited. Some references (eg reference 33) are incomplete.
- Table 1 misses the characteristics of compounds 4b-m and the interpretation of ESOL, GI and BBB.
- For all compounds, the formula of [M + H] + have one less proton number, the calculated exact mass are incorrect.
Author Response
We appreciate the helpful suggestions and constructive comments provided by the reviewers. Furthermore, we are pleased that the reviewers found the article interesting and worthy of publication. We have made major revisions as requested. We have revised the manuscript in response to the reviewers’ comments and believe that our revisions have significantly improved the quality of this manuscript. We have shown the changes in the manuscript with yellow highlights. Below please find our point-by-point response to the comments of the reviewers, as shown in blue arial font.
Reviewer #1:
This manuscript describes the design, synthesis, and biological evaluation of chrysin-de-allyl PAC-1 hybrid compounds as anticancer candidates. Although the design and synthesis methods are not novel, and the biological activity of the hybrid compounds is not more potent than that in previous works, but half of novel hybrid compounds have moderate antiproliferative activity against MDA-MB-231, especially, 4e and 4g have higher cytotoxicity to cancer cells than normal cell lines. In addition, the authors also conducted flow cytometry and protein expression studies on 4e and 4g and confirmed that intrinsic apoptosis leads to cytotoxicity. These results and conclusions will provide Molecules’ readers with interesting information. Therefore, I recommend accepting it for publication. However, the author needs to address some issues."
Response:We thank the reviewer for their critical review of this manuscript. We also appreciate that the reviewer find the work interesting for Molecules audience.
Comment #1:Based on Figure 4, compound 4e does not appear to activate Bak in a concentration-dependent manner.
Response:Thank you for your observation. (Please note that 4e became 4g and 4g became 4i in the revised manuscript based on other reviewer corrections). We agree that the apoptotic effect of 4g and 4i are not concentration - dependent. For compound 4g, only the expression of cleaved caspase 7 and caspase 9 are concentration – dependent, whereas the release of cytochrome c is significantly increased only by the 5 µM concentration (p<0.05). Similarly, for compound 4i, only caspase 9 is significantly activated in a concentration - dependent manner, whereas the expression of most of the proteins, such as caspase 3, cleaved caspase 9 and cytochrome c, are significantly increased only at the highest concentration (10 µM), with caspase 7 being significantly activated only by the 5 µM concentration of 4i. We acknowledge that more work needs to be done to understand this phenomenon in future studies. In the revised manuscript, we have removed the term “concentration-dependent manner” in the results and conclusion section as most of the proteins were activated at a single high concentration. Adding more concentrations on the higher side at different time-points may give us more clarity in future studies.
Comment #2: References 31 and 31 are improperly cited. Some references (eg reference 33) are incomplete.
Response:Thank you for noticing this mistake. All referenecs are proof-checked and incomplete references were corrected. Please see the references on pages 18-21.
Comment #3:Table 1 misses the characteristics of compounds 4b-m and the interpretation of ESOL, GI and BBB.
Response:To resolve this problem, we have added interpretations to all of the parameters of Table 1. We have also added additional characteristics such as lead-likeness and Log P Values as parameters to Table 1, as we felt that it was crucial for characterizing the listed compounds.
Comment #4:For all compounds, the formula of [M + H]+ have one less proton number, the calculated exact mass are incorrect.
Response:We appreciate this comment. In the original manuscript, the molecular formulas for all compounds were reported in the unionized forms and the exact masses were reported for the ionized forms of the corresponding compounds. To resolve this discrepancy, the molecular formulas for all compounds in the revised manuscript were changed to the ionized forms (M+H)+. Please see track changes in the revised manuscript in pages 14-16.

Reviewer 2 Report
Authors suggest that compounds 4e and 4g induced apoptosis by 1) inducing 27 cell cycle arrest at the G2 phase in MDA-MB-231 cells and 2) activating the intrinsic apoptotic 28 pathways in a concentration-dependent manner among chrysin-De-allyl-Pac-1 hybrid analogues. Despite interesting data, it has some concerns as follows:
- Generally it is well organized and written. But add bar graph for Fig 4. Apoptotic effect of 4e and 4g is not in a concentration dependent?? Discuss it. Especially cytochrome c, caspase9
- Apoptotic effect looks different by 4e and 4g, implying the inconsistency of data in Fig3 and Fig4
- Hence, you have to show more convincing data. How about their effect on PARP, caspase 8
- Show caspase inhibitor study on cytotoxicity, sub G1, Caspase 3 and PARP
- Check grammar flaws; TNBC often present
Author Response
We appreciate the helpful suggestions and constructive comments provided by the reviewers. Furthermore, we are pleased that the reviewers found the article interesting and worthy of publication. We have made major revisions as requested. We have revised the manuscript in response to the reviewers’ comments and believe that our revisions have significantly improved the quality of this manuscript. We have shown the changes in the manuscript with yellow highlights. Below please find our point-by-point response to the comments of the reviewers, as shown in blue arial font.
Reviewer #2:
"Authors suggest that compounds 4e and 4g induced apoptosis by 1) inducing 27 cell cycle arrest at the G2 phase in MDA-MB-231 cells and 2) activating the intrinsic apoptotic 28 pathways in a concentration-dependent manner among chrysin-De-allyl-Pac-1 hybrid analogues. Despite interesting data, it has some concerns as follows:"
Comment #1:Generally, it is well organized and written. But add bar graph for Fig 4. Apoptotic effect of 4e and 4g is not in a concentration dependent?? Discuss it. Especially cytochrome c, caspase9
Response:We thank the reviewer for indicating that our study was well designed, organized and well written and the data were interesting. We also thank the reviewer for their critical evaluation that has improved our study. (Please note that 4e became 4g and 4g became 4i in the revised manuscript based on reviewer corrections). We agree that the apoptotic effect of 4g and 4i are not concentration - dependent. For compound 4g, only the expression of cleaved caspase 7 and caspase 9 are concentration – dependent, whereas the release of cytochrome c is significantly increased only by the 5 µM concentration (p<0.05). Similarly, for compound 4i, only caspase 9 is significantly activated in a concentration - dependent manner, whereas the expression of most of the proteins, such as caspase 3, cleaved caspase 9 and cytochrome c, are significantly increased only at the highest concentration (10 µM), with caspase 7 being significantly activated only by the 5 µM concentration of 4i. We acknowledge that more work needs to be done to understand this phenomenon in future studies. In the revised manuscript, we have removed the term “concentration-dependent manner” in the results and conclusion section as most of the proteins were activated at a single high concentration. Adding more concentrations on the higher side at different time-points may give us more clarity in future studies.
Comment #2:Apoptotic effect looks different by 4e and 4g, implying the inconsistency of data in Fig 3 and Fig 4
Response:The apoptotic effect of compound 4g and 4i are similar based on morphological data and biochemical studies. Both of these compounds had the apoptotic phenotype in MDA-MB-231 cells at 20 µM of 4g and 4i. The number of adherent MDA-MB-231 cells decreased in number and exhibited cellular shrinkage. The cells were rounded, loosely attached and apoptotic bodies were present. Both of the compounds, 4g at 5 µM and compound 4i at 10 µM produced a significant increase in the expression of cytochrome c, compared to cells incubated with vehicle, followed by caspase 9 activation and subsequent activation of the executioner caspase, caspase 7. Please see the revised figure 3. We have also added new morphological data and figures, that was not previously presented. Figure 3 and 4 are now revised.
Comment #3, 4:Hence, you have to show more convincing data. How about their effect on PARP, caspase 8. Show caspase inhibitor study on cytotoxicity, sub G1, Caspase 3 and PARP
Response:We agree with the reviewer that the effect of these compounds on PARP and caspase 8, as well as the effect of caspase inhibitor on cytotoxicity, will provide more robust data to delineate the mechanism of cell death induced by the lead compounds. However, due to COVID-19, research operations at the university is suspended and since these experiments require lab work, we will conduct such experiments in the future. Thank you for understanding.
Comment #5:Check grammar flaws; TNBC often present
Response:The manuscript was proofed for grammar. Limited use of TNBC is now indicated in the revised manuscript. Thanks

Reviewer 3 Report
The manuscript presents the design, synthesis and biological evaluation of chrysin-De-allyl PAC-1 hybrid analogues as anticancer compounds. There is general interest in the development of novel molecules to treat triple negative breast cancer. The reports builds on a similar study that was published last year by the same authors (Abdallah Al-Oudat, B., et al., Design, synthesis, and biologic evaluation of novel chrysin derivatives as 534 cytotoxic agents and caspase-3/7 activators). In the published paper they developed a series of compounds as chrysin derivatives (3a-m). In the submitted manuscript they show a new series and refer to them as series 4, in which the compounds 4e and 4g exhibited the highest efficiency against triple negative breast cancer cells. It is interesting that a comparison of the IC50 values for two compounds reported in the previous published work that are very similar in terms of chemical structure to compounds reported in the manuscript submitted to Molecules showed very different IC50 values against the triple negative breast cancer cell lines MDA-MB-231. To mention two examples, the IC50 for compound 3c and 4a are 54.8±5.7 and 6.8 ± 2.76, respectively. The only difference is the position of the OH functional group in the aromatic ring of the R-substituent. Similarly, the IC50 for compound 3d and 4c are 3.3±0.8 and 23.3 ± 6.98, respectively, being the position of the methoxy group the only difference between the two molecules. It will be important to discuss more thoroughly these differences in relation to the compounds biological activities. It would have been very interesting to include the compound 3c of the previous report in the current study to see more clearly the effect of the OH group position in the aromatic ring (in addition to polyhydroxylation) on the compounds anticancer activity. I encourage the authors to consider the inclusion of such data to enhance the impact of this contribution.
Author Response
We appreciate the helpful suggestions and constructive comments provided by the reviewers. Furthermore, we are pleased that the reviewers found the article interesting and worthy of publication. We have made major revisions as requested. We have revised the manuscript in response to the reviewers’ comments and believe that our revisions have significantly improved the quality of this manuscript. We have shown the changes in the manuscript with yellow highlights. Below please find our point-by-point response to the comments of the reviewers, as shown in blue arial font.
Response:We thank the reviewer for these comments. We have extensively revised the manuscript based on the reviewer suggestions. The table shown below summarizes the IC50values for the four compounds mentioned above. Please note that the IC50value for compound 3d is 11.9 ± 3.5 µM not 3.3 ± 0.8 µM [1]. The difference in the efficacy of the compounds is perhaps due to chemical modifications either by alkylation or shift in the position of the hydroxyl group. This is not an unexpected outcome as often small modifications in the chemical structures of the compounds may lead to large differences in the corresponding biological efficacies. Furthermore, it should be noted that often times the cell-cytotoxicity assays like MTT can provide small shift in IC50s. The IC50currently reported was verified by the data from Incucyte and colony formation assay and is repeatable. We have provided the details in the revised manuscript. Thank you.
|
This work |
IC50 (µM) |
|
Previous work[1] |
IC50 (µM) |
4a |
6.8 ± 2.76 |
3C |
54.8 ± 5.7 |
||
4c |
23.3 ± 6.98, |
3d |
11.9 ± 3.5 |
||
|
|
|
|
|
|
- Abdallah Al-Oudat, B., et al., Design, synthesis, and biologic evaluation of novel chrysin derivatives as cytotoxic agents and caspase-3/7 activators.2019. 13: p. 423-433.

Round 2
Reviewer 2 Report
Much improved
Author Response
Thank you. I appreciate the comments. The English are again proofed. And we did find some minor corrections. See the track change version.
Again, I thank the reviewer for carefully looking at the manuscript the second time.
Reviewer 3 Report
There are a few typo/grammar issues in the revised manuscript These are listed as follows:
Line 212: A full stop is missing after the word cycle.
Line 213. For the sake of consistency, use ; after the word respectively.
Line 243> Remove the extra space between the words in breast.
Lines 255-256. Please amend the grammar "where Clvd is for
256 cleaved".
Author Response
We thank the reviewer for noticing the typos and grammatical errors. This is now been proofed carefully. All the corrections mentioned by the reviewer are fixed.
Thank you once again for reviewing our work.
Sincerely yours,
Amit K. Tiwari